# The Role of the Extracellular Matrix (ECM) in Wound Healing: A Review

**DOI:** 10.3390/biomimetics7030087

**Published:** 2022-07-01

**Authors:** Robert B. Diller, Aaron J. Tabor

**Affiliations:** Department of Biological Sciences, Northern Arizona University, Flagstaff, AZ 86011, USA

**Keywords:** wound healing, extracellular matrix (ECM), inflammation, fibroblast(s), collagen, full-thickness wound, dermal mimics, hemostasis, proliferation, tissue remodeling, granulation, first-intention healing, second-intention healing

## Abstract

The extracellular matrix (ECM) is a 3-dimensional structure and an essential component in all human tissues. It is comprised of varying proteins, including collagens, elastin, and smaller quantities of structural proteins. Studies have demonstrated the ECM aids in cellular adherence, tissue anchoring, cellular signaling, and recruitment of cells. During times of integumentary injury or damage, either acute or chronic, the ECM is damaged. Through a series of overlapping events called the wound healing phases—hemostasis, inflammation, proliferation, and remodeling—the ECM is synthesized and ideally returned to its native state. This article synthesizes current and historical literature to demonstrate the involvement of the ECM in the varying phases of the wound healing cascade.

## 1. Introduction

Wound healing is a complex and coordinated cascade of events that helps to maintain homeostasis within the integument [1,2] and subsequently protect the whole organism [3]. When considering dermal wound healing, the longer a wound takes to heal, the greater the opportunity for foreign agents to enter the body and have pathological effects. To help mitigate the duration of the wound, implanting materials into the integumentary can alter the wound healing cascade of events [4,5]. Currently, translational scientists are researching and developing novel ways to create bio-inspired scaffolds able to reproduce the extracellular environment of the native tissues [6,7]. It has been estimated that the advanced wound care market will be approximately 19 billion dollars by 2027 [8]. Having these technological advancements offers patients within this growing market novel treatment regimens many of which will involve reconstruction of the native ECM. Granted, the type of wound would be the greatest indicator of the need to treat with the addition of an implanted bio-inspired scaffold.

The etiologic agent will dictate the type of wound healing that occurs. First intention wound healing is often a result of surgical incisions or a clean laceration [9]. These first intention wounds do not lead to fibrotic formation rather heal through connective tissue and epithelial regeneration [9,10,11]. Second intention wound healing is a more involved process and often occurs during soft tissue loss such as ulcerations, severe burns, and major surgeries [9,10]. Second intention healing occurs through the formation of granulation tissue which is then followed by the synthesis of extracellular matrix (ECM), much of which is due to fibroblasts [9]. These second intention wounds close through wound contraction by way of myofibroblasts and re-epithelialization from the wound margins [12,13]. Second intention (full thickness) wounds healing time is often delayed due to infection or disease states such as diabetes, which alter the normal physiological processes [9,14,15,16]. During healing by second intention, the protein elastin, responsible for the elastic recoil of the dermal matrix, is absent from the healing granulation tissue deposited by resident fibroblasts [17]. The lack of elastin creates a more rigid and inelastic ECM. Collagen, which provides tensile strength to the skin, is also secreted into the healing wound space [17]. The secreted collagen is disorganized, lacks structural integrity, and the resulting scar tissue never achieves the tensile strength of native skin [17]. Current methods to treat intractable wounds that involve second intention healing include the use of antibacterial, debridement, irrigation, vacuum-assisted closure, oxygenation, moist wound healing, hirudotherapy, and collagen-based dressings/fillers [18,19,20,21,22].

Restoration of a dermal matrix that mimics the unwounded structure and function, preserving the dermal integrity, would theoretically improve the resulting scar so that the tensile strength and elastic recoil would approximate that of unwounded, intact dermis [17]. The microarchitecture of scaffolds is relevant to tissue engineering due to the ability of the scaffolds to mimic native ECM in scale, which is thought to encourage cellular ingrowth, ECM deposition, and neotissue formation [23,24,25,26]. Table 1 contains a list of products used for wound healing which utilize ECM proteins, and a short list of medical devices created from synthetic polymers which are manufactured to mimic the ECM.

## 2. Involvement of Extracellular Matrix in Normal Wound Healing

Normal wound healing is a dynamic process that involves epithelial and dermal regeneration as well as scar development, which includes ECM regeneration. When normal tissue is disrupted, either cut surgically or through ulcerations, a healthy organism must be able to repair itself for longevity and survival. The normal wound healing model is characterized by four phases: clotting and coagulation (hemostasis), inflammation, proliferation, and remodeling [9,43,44,45,46]. These phases are not mutually exclusive, overlap in the organism (Figure 1), and typically occur quickly (e.g., days to weeks), while the remodeling phase of the wound healing responses can take up to months (12 months) for completion [45,47].

The ECM is an acellular, protein-rich matric that is essential for structural support and cellular attachment [48,49,50,51]. Previously published studies have supported the ECM in modulating wound healing by regulating biochemical pathways and biomechanical signaling pathways [49,51,52,53,54]. The ECM directly modulates aspects of cell behavior, including adhesion, proliferation, migration, and survival [12,54,55,56]. Indirect modulation involves cells acting on the ECM stimulates extracellular protease secretion and modulating growth factor bioavailability [54].

The two most ubiquitous proteins found in the native ECM are collagen and elastin. Collagen is the most prevalent protein found in the ECM [57,58]. It comprises approximately 50–90% of the integument [19,59]. Table 2 details the percent composition of the ECM. Collagen is predominately synthesized by fibroblasts and there are many variants of collagen. Collagen type I is the dominant form in adults, while collagen type III is the most abundant during gestational development [60]. Interestingly, in 2020 it was published that the precursor to collagen, pre-collagen, is regulated by the 24-h circadian cycle [61]. Elastin, on the other hand, comprises 0.6–7.9% of the dermal ECM depending on anatomical location, gender, and age [59,62,63,64,65,66,67]. Other proteins found in the ECM include laminin, fibronectin, vitronectin, and tenascin, as well as proteoglycans and glycosaminoglycans (GAGs) that make up the remaining 3–5% of the ECM [68,69]. For the most part these proteins will be created in the newly formed ECM, with the exception of elastin. In human skin and most organs, the bulk of elastogenesis occurs during late fetal and early neonatal periods. By maturity production it is complete and synthesis of new tropoelastin ceases [67,70,71]. Elastin essentially does not turn over in healthy tissues, therefore fiber function and tissue integrity are compromised due to this limited pattern of elastin production over the life of specific organs [70,72,73,74].

## 3. Clotting and Coagulation (Hemostasis)

Wound healing begins immediately following insult to the tissue when platelets contact exposed collagen from the endothelial wall and ECM initiating clotting and coagulation [75,76,77,78]. Upon adhesion to collagen, adenosine diphosphate (ADP) is released from platelet granules and erythrocytes to initiate platelet aggregation. An anti-hemophilic factor (Factor VIII) is also released from the platelet alpha granules [79,80]. Factor VIII accelerates the formation of the platelet plug and, in conjunction with the fibrin network, fills the damaged tissue space. This structure provides a scaffold for cellular migration and proliferation, essentially a provisional extracellular matrix (ECM) [12,38,81,82]. The fibrin network is rapidly degraded by plasmin and neutrophil elastase, which can release the plasma growth factors trapped in the fibrin clot [38,83]. The fibrin degradation products also stimulate ECM deposition [84], fibroblast proliferation [85], and angiogenesis [86].

The recruitment of cells to the damaged tissue area is an important component of wound healing [44]. This mechanism primarily relies on two growth factors: platelet-derived growth factor (PDGF) and transforming growth factor β (TGF-β) [68,77,87,88,89]. PDGF and TGF-β initiate chemotaxis of neutrophils, macrophages, fibroblasts, and smooth muscle cells. These processes are required for the wound healing cascade to progress to the next stage, inflammation [77,90,91].

## 4. Inflammation

Following platelet activation, the first inflammatory cells to invade the wound site are the neutrophils, which are recruited to the site by PDGF and TGF-β [12]. These neutrophils aid in protection against infection and removal of tissue debris. Following neutrophils, monocytes and T-lymphocytes occupy the tissue. The recruitment of monocytes and T-lymphocytes are the first overlap of steps between hemostasis and inflammation. The fragmentation of fibronectin and other ECM components act as chemoattractants for monocytes which bind ECM proteins [17,92]. This binding stimulates phagocytosis, [17,93] leading the monocytes/macrophages to further break down ECM fragments and other debris in the area [17,91,94]. Adherence of monocytes to ECM proteins also stimulates the expression of growth factors [17,95] that can then act on cells to affect the synthesis of ECM components (e.g., proteoglycan synthesis by fibroblasts) [17,96,97]. Inflammation plays a primary role in prevention of infection of the tissue [98,99].

Acute inflammation initiates within 24 h of injury and is characterized by initial vasoconstriction followed by vasodilation and increased capillary permeability [100]. Neutrophils are recruited to the wound site within an hour by factors released from the platelets in the hemostasis phase (including interleukin-1, TNF-α, TGF-β, and platelet factor-4,) and have a sustained migration for 48 h [78,101,102]. Neutrophils from the circulating blood are recruited through molecular changes on the surface of endothelial cells [103].

The primary function of infiltrating neutrophils is to remove foreign or damaged particles, bacteria, and non-functioning host cells found in the wound [104]. Neutrophils accomplish this through phagocytosis, degranulation, and the production of chromatin and protease traps [98]. Granules release several toxic enzymes that include lactoferrin (antibacterial effect), proteases, neutrophil elastase, and cathepsin [54,98]. The resulting debris either becomes part of the scab and is sloughed off or phagocytized by macrophages [100,101]. At the same time neutrophils are recruited, a circulating monocyte influx is stimulated by the fragmentation of fibronectin [102], elastin [103], collagen [104], enzymatically active thrombin [105], complement components, PDGF, TGF-β, leukotriene B_4_, and PF-4 [10]. Infiltrating monocytes become activated macrophages and function to phagocytize effete neutrophils and residual bacterial particles at the wound site. Activated macrophages release growth factors such as PDGF, vascular endothelial growth factor (VEGF) and cytokines that are required for the formation of granulation tissue and fibroblast proliferation [77,81,106,107]. Macrophages continue cleaning the wound site and stay in the wound site much longer [107,108,109].

The final cells recruited during inflammation are lymphocytes. These cells are attracted by several chemoattractants including IL-1, complement components and immunoglobulin G (IgG) breakdown products [98,100,107]. The lymphocytes regulate the proliferative phase of wound healing [98,107,110]. The role of T-lymphocytes is not completely understood and is a current area of research [2].

Another important cell in the inflammation process is the mast cell [111,112,113,114]. Mast cells release granules of histamine, heparin, interleukins (IL-6, IL8), VEGF, enzymes, and other active amines which together are responsible increasing vascular permeability and the characteristic signs of inflammation, calor (heat), tumor (swelling), rubor (redness), and dolor (pain), that surrounds the wound site [111,113,114]. The proteins released from these granules cause the surrounding vessels to become more permeable, which allows mononuclear cells and fluid to pass into the wound area giving way to the signs of inflammation [114,115].

Inflammatory resolution is largely controlled by anti-inflammatory interleukins (IL-10) and the genetic regulation of inflammatory genes [116]. During this down regulating event, additional cells will migrate to the injured tissue site. These cells include mesenchymal stem cells, fibroblasts that create extracellular matrix, and nearby endothelial cells that create new blood vessels (i.e., angiogenesis) [117]. At the end of the inflammatory phase, a rich granulation tissue has begun to be synthesized that contains a significant number of fibroblasts, capillaries, and inflammatory cells [117,118].

## 5. Proliferation

The next stage of wound healing is proliferation, which is marked by the rapid formation of new connective tissue that functions to repopulate the wound bed with a newly formed ECM. However, during adult wound healing the newly formed ECM does not resemble the native ECM prior to the injury [17]. The ECM during the adult wound healing event is much looser which allows for cellular invasion [81]. This stage is termed granulation tissue formation due to the gross granular appearance of the tissue when excised from the wound [43]. The tissue has a granular appearance due to the large amount of neovascularization (new blood vessels) that has taken place [43,117]. Figure 2 below details granulation tissue in a chronic wound.

During proliferation, granulation tissue converts from a cell-rich, highly vascular medium to relatively avascular and acellular matrix of collagen [119,120]. However, elastin is absent from the formed granulation tissue deposited by resident fibroblasts [17]. Elastin has only been demonstrated to be present after these wound stages have progressed, in some cases months after this proliferative event [53,67,121]. Instead, collagen provides tensile strength to the skin and is mostly secreted into the healing wound space in a disorganized way, lacking structural integrity [17,122].

After approximately 96 h, the proliferation phase initiates, and new stroma begin to invade the wound site [82]. Figure 3 details the phases of wound healing, including proliferation. Granulation tissue consists of a dense population of macrophages, fibroblasts, and neovasculature embedded in a loose matrix of collagen, fibronectin, and hyaluronic acid [123,124]. All these cells and proteins move into or are deposited into the wound space at about the same time [81]. One such cell type is the fibroblast which are signaled into the wound bed by following the collagen protein orientation that has been deposited [120,125]. As fibroblasts invade the fibrin clot, the cells are lysed and then deposit hyaluronan and fibronectin; this forms the early granulation tissue [117]. Hyaluronic acid (HA) is a large component of early granulation tissue [126,127]. The process of granulation initially occurs in the periphery of the clot and moves toward the center as the granulation tissues grow into the wound space [117,127]. The ECM of the healing wound undergoes rapid changes as the fibrin clot is replaced by fibronectin and hyaluronan and subsequently by collagen types I and II [43,117,128].

The newly forming vessels that arise during neovascularization sprout from existing vessels that are stimulated by the cytokine VEGF, fibroblast growth factor (FGF)-2, PDGF, and members of the TGF-β family [127,129,130]. Small capillary projections are initially formed through the accumulation of endothelial cells on the leading edge, with further extension occurring from endothelial proliferation [131]. The maturation of these vessels occurs with the emergence of peri-endothelial cells that aid in matrix formation and undergo a change to become pericytes (contractile cells that wrap around endothelial cells) [132]. This process is perpetuated by recruited macrophages, which continue to provide the cytokines necessary for fibroplasia.

The persistence of fibroblasts produces a new extracellular matrix that supports new cellular growth [117]. The accumulation of fibroblasts does not necessarily ensure that collagen will be synthesized and deposited; the fibroblasts must first be provided with a stimulus. This stimulus comes from the production of lactate synthesized by metabolically active macrophages and new blood vessels [133,134,135]. The neovasculature must be present to provide oxygen and nutrients to support cellular metabolism [81]. Along with neovascularization, collagen production, macrophage activation, and epithelialization occur in the proliferating tissue.

Epithelialization is stimulated by the presence of epidermal growth factors and TGF-α. These growth factors are produced by activated macrophages, platelets, and keratinocytes [136,137]. For the epidermal cells to differentiate and re-establish the permeability barrier, fibroblasts in the granular tissue must transform into myofibroblasts that contract the wound [43,124]. Once the new epithelial cells span the wound site, enzymes are released to allow the scab to be removed from the damaged tissue. The result of the proliferative phase is the increased presence of new vessels and fibrous tissue that will provide the basic foundation for tissue remodeling. Throughout the granulation tissue stage, matrix formation and remodeling occur simultaneously and overlap considerably.

## 6. Remodeling

The final phase of wound healing is tissue remodeling. Tissue remodeling allows for the restoration of the tissue architecture by macrophages and fibroblasts, which helps to restore tissue strength [77,96]. The length of this phase is highly variable, many times related to the wound size and pre-existing conditions of the patient. On average, this phase begins two to three weeks after the initial tissue injury event and can take up to a year or longer for remodeling to occur [138]. The length of remodeling is predominately due to the synthesis and degradation of collagen in the ECM of the wound bed [98]. During the remodeling period of wound healing, the fibronectin-rich matrix is replaced with a stronger collagenous matrix [139]. TGF-β1 stimulates production of collagen type I and III in fibroblasts [44,138]. This collagenous matrix is strengthened due to the crosslinking and structural modifications it undergoes throughout the remodeling phase. This crosslinking is completed by varying enzymes including transglutaminases and lysyl oxidases [140]. Wound resolution begins as the amount of extracellular collagen increases and fibroblasts decrease their collagen production [43,141]. During the course of a year, type III collagen is replaced by type I collagen through regulation by matrix metalloproteinases (MMPs) [142,143]. There are 24 distinct extracellular endopeptidases in the MMP family [144,145]. MMPs are proteases that breakdown ECM proteins which modulate the ECM environment [146]. The result of normal healing is a type I collagenous scar that is largely avascular and lacks any ordered structure. This is exemplified by the fact that healed wounds never regain full tensile strength [147,148]. On average, the wounds reach 50% tensile strength at three (3) months and the maximum tensile strength a wound can achieve is approximately 70–80% of normal tissue [16,43,77,100,133].

## 7. Conclusions

The extracellular matrix (ECM) is a critical structural component of any tissue, including the integument. Damage to the ECM through an acute or chronic source can lead to a series of healing events, coined the “wound healing cascade”. These overlapping events begin with the initial cessation of blood flow and clotting called hemostasis, followed by the inflammatory stage, progressing to the granulation-rich proliferative stage and finally the remodeling phase, which produces a collagen I tissue. All of this is responsible from cellular signaling events, many of which involve the ECM. This review addresses the role of the ECM throughout the wound healing cascade of events.

Many wound healing products currently on the market in the US have realized that using these structural proteins can increase the rate of wound heling in second intention wounds. However, this may not be enough. It is the opinion of the authors that more research needs to be devoted to not only utilizing the structural components of the ECM but to also mimic the physiological structure.

## Figures and Tables

**Figure 1 biomimetics-07-00087-f001:**
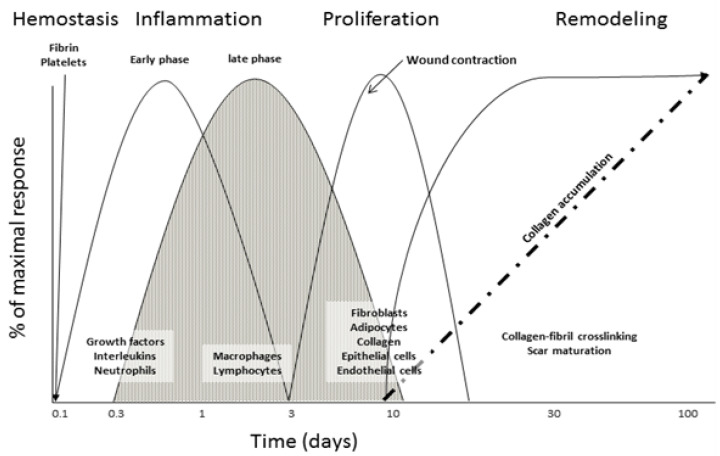
Schematic of the classical wound healing cascade with important stages of cellular infiltration and protein deposition.

**Figure 2 biomimetics-07-00087-f002:**
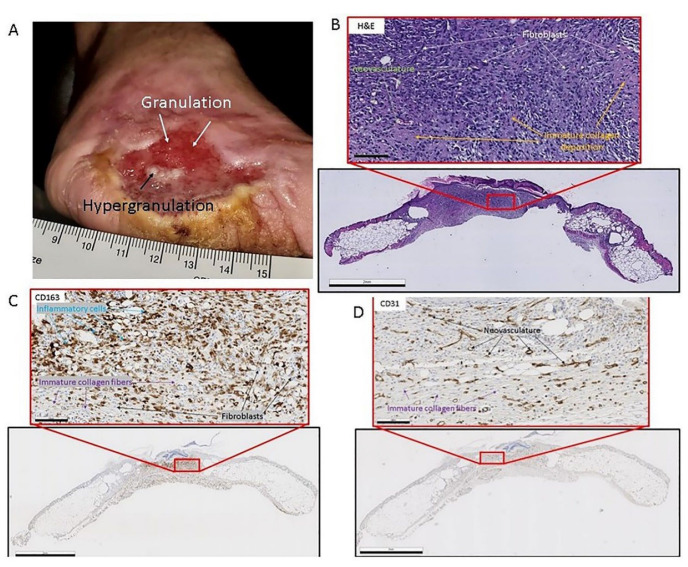
(**A**) Clinical image of chronic diabetic injury with granulation tissue presence. The wound is on the medial aspect of the foot. Notice the white arrow pointing to red tissue indicating cellular infiltrate and neovascularization. The black arrow is pointing to the area in the middle of the wound which is indicative of hyper-granulation. (Patient consent provided). (**B**) Heamatoxalin and Eosin-stained murine skin the black frame is a gross image (scale bar = 2 mm) of the skin sample, and the red framed image (scale bar = 100 µm) is a magnified image of the area in the small red box. The white arrows point to fibroblasts in the granulation tissue. The orange arrows point to immature collagen deposition. The green arrows point to neovasculature. (**C**) CD-163 (Inflammatory cell marker) reacted immunohistochemistry-stained murine skin the black frame is a gross image (scale bar = 2 mm) of the skin sample and the red framed image (scale bar = 100 µm) is a magnified image of the area in the small red box. The black arrows point to fibroblasts in the granulation tissue. The purple arrows point to immature collagen deposition. The blue arrows point to inflammatory cells: macrophages, and monocytes. (**D**) CD-31 (endothelial cell marker) reacted immunohistochemistry-stained murine skin, the black frame is a gross image (scale bar = 2 mm) of the skin sample, and the red framed image (scale bar = 100 µm) is a magnified image of the area in the small red box. The black arrows point to neovasculature in the granulation tissue. The purple arrows point to immature collagen deposition.

**Figure 3 biomimetics-07-00087-f003:**
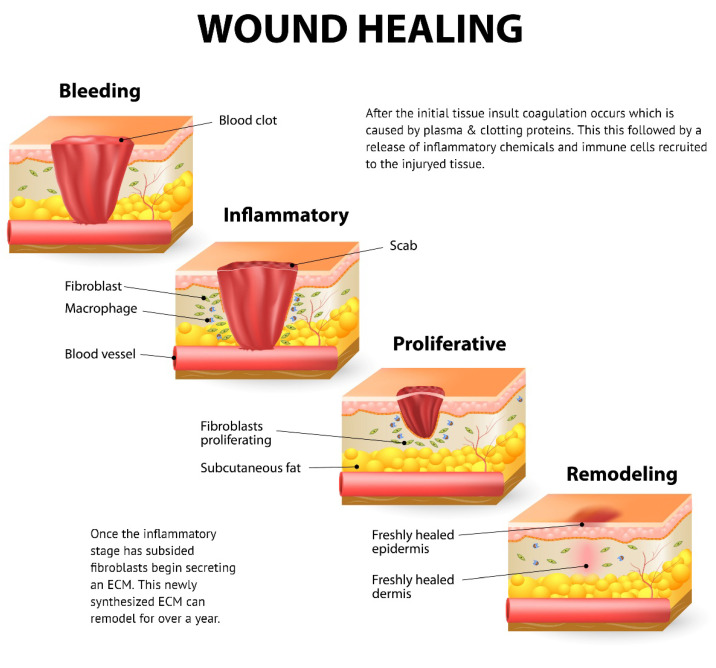
Phases of the wound healing response including hemostasis (coagulation), inflammation, proliferation, and tissue remodeling (Designua/Shutterstock.com).

**Table 1 biomimetics-07-00087-t001:** A list of wound healing products created using ECM proteins and a list of medical devices manufactured from synthetic polymers, intended to mimic the ECM. These lists are not meant to be all inclusive, but instead provided the reader with several examples of products which utilize ECM proteins to assist in wound healing, and products which attempt to mimic the structure of the dermis to influence wound healing.

Products Manufactured Utilizing ECM Proteins				
Product Name	Source	Composition	ECM Proteins	Reference(s)
Acell Wound Powder	Porcine Urinary Bladder	Powdered Urinary Bladder Extracellular Matrix	Collagen I, Collagen III, and Collagen IV	[27]
AlloDerm	Human Dermis	Human acellular, lyophilized dermal matrix	collagen, elastin, basement membrane, hyaluronic acid glycosaminoglycan (GAG)	[28,29,30]
AlloPatch	Human Cadaveric Dermis	aseptically processed human reticular dermal tissue	collagen, elastin, basement membrane, hyaluronic acid glycosaminoglycan (GAG)	[29]
Axolotl DualGraft	Human Amnion	Dehydrated amniotic membrane	Collagen I, Collagen II, Collagen IV, Collagen V, Collagen VI, Proteoglycans, Fibronectin, Laminin	[31]
GraftJacket	Human cadaveric skin	Cryogenicaly preserved decellularized human dermis	Collagen and elastin	[32,33]
MatriDerm	Bovine Ligamentum nuchae	3D matrix of type 1 collagen fiber coated with a 3% a-elastin hydrolysate	collagen fibrils types I, III, and V and alpha-elastin	[34,35]
OASIS	Porcine jejunum submucosa (PSIS)	Lyophilized small intestine submucosa	Collagen I, Collagen III, Collagen IV, Collagen VI, fibronectin, elastin, hyaluronan, chondroitin sulfate, decorin	[36,37]
Pelnac	Porcine Achilles tendon	Collagen sponge porous matrix	Collagen	[34]
PalinGen Membrane	Human Amnion	Dehydrated amniotic membrane	Collagen I, Collagen II, Collagen IV, Collagen V, Collagen VI, Proteoglycans, Fibronectin, Laminin	[31]
Permacol	Porcine Dermis	Collagen and Elastin crosslinked by diisocynate	Collagen I, Elastin	[38]
Promogran	Bovine Dermis	55% Bovine Dermal Collagen 45% Oxidized regenerated cellulose	Collagen	[39]
SimliDerm	Human Cadaveric Dermis	pre-hydrated human acellular dermal matrix	Collagen, elastin, basement membrane, hyaluronic acid glycosaminoglycan (GAG)	[30]
Xcellistem	Porcine Spleen and Lung	wound powder composed of a blend of multiple porcine-based extracellular matrix material	Collagen I, Collagen III, Collagen IV, Sulfated glycosaminoglycan, Hyaluronic acid, Lipids, Elastin, Fibronectin, and Laminin	[40]
Novosorb	Synthetic polyers	polyurethane foam made from ethyl lysine diisocyanate, lactic acid/ethylene glycol chain extender, and PCL1000 polyol with a removable polyurethane overlayer	None	[41]
Phoenix Wound Matrix	Sythetic polymers	Electrospun nonwoven fibrous three-dimensional matrix comprised of Poly(lactide-co-caprolactone) and Polyglycolic acid fibers	None	[42]
Restrata Wound Matrix	Synthetic polymers	electrospun mat of polyglactin 910 (PGLA) and polydiaxonone (PDO) fibers	None	[41]

**Table 2 biomimetics-07-00087-t002:** Listing of proteins found in ECM, percent composition, and the general functions within the ECM of the integument. Note that varying degrees of composition can occur because of the location of ECM.

Protein	Percent Composition	Function(s)
Collagen(s)	50–90	Synthesized by fibroblasts. Gives structural and tissue integrity, aids in epidermal/dermal differentiation.
Elastin	0.6–7.9	Creates an intricate network for structural support allowing for elasticity of tissue.
Fibronectin	<1.0	Involved in wound healing including platelet spreading and leukocyte migration to injured tissue(s). Aids in promotion of elastin deposition and mechanical strength of ECM.
Laminin	<1.0	A glycoprotein that is a part of the basal lamina, aids in cellular signaling.
Vitronectin	<1.0	A glycoprotein involved in hemostasis and cellular adhesion during tissue damage.
Tenascin	<1.0	Glycoprotein family that aids in cellular migration adhesions and cell proliferation.

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
