# Peer review of "The Role of the Extracellular Matrix (ECM) in Wound Healing: A Review"

_biomimetics, 2022, doi:10.3390/biomimetics7030087_

Round 1

Reviewer 1 Report

This is a manuscript written by Robert B Diller and Aaron J. Tabor, reviewing the role of the ECM in wound healing. The manuscript is nicely written, I just put several comments to improve the quality of the manuscript.

1- The prepared figures are mostly clear and nicely presented. It would be good to refer to them more in the text and elaborate on them in the according sections in the text.

2. Please clearly mention the novelty of your review. It seems to me that the novelty of this work is not enough for publication.

3. Make sure if you have permission to re-use the figures.

4. Figure 2 is not clear and is too small as a figure. Please add other parts to this figure to be more informative.

5. A table summarizing the process can be appreciated.

6. Please discuss the challenges and future perspective for ECM role in wound healing. What would be the application of ECM in the lab? Tissue engineering? 

7. The references are really outdated. Please use the recent publications.

Author Response

Reviewer One:

Dear Reviewer,

Thank you for taking the time to review our submission. In response to your comments please note the following:

  • As per your note we have updated and elaborated on the images, along with adding histological representation that adds o the explanation.
  • As for the novelty, this was intended to be a review of the ECM in its role in wound healing, while we agree that the review did not include our new research the goal was to review both historical and more recent publications which have been added to the article.
  • All figures are created by the authors, or we own/have permission to use. Thank you for checking on this.
  • Figure 2 has been updated to be more informative, including a greater description and addition of gross and microscopic (histology) representation.  
  • Tables including the current marketed /ECM products available and another with the ECM protein composition.
  • Additional references have been added including 2020 and newer.

We welcome any additional critiques to improve the review article.

Reviewer 2 Report

In this review article, the authors discuss the role of the extracellular matrix (ECM) in wound healing.

The ECM, in addition to its structural role, also provides instructive cues. During wound healing, the skin develops scar tissue characterized by dense and fibrotic ECM with altered structure and function, costing more than $20 billion in the US alone. This review summarizes recent developments in wound healing and the role of ECM, and this will attract many readers from diverse fields.

I would recommend this review be accepted in the present condition. However, since fibroblasts are considered important in secreting and organizing ECM, recent work published on fibroblast Engrailed-1 (Mascharak et al., Science 372, eaba2374 (2021) 23 April 2021) can enhance the review.

Best

Author Response

Reviewer Two:

Dear Reviewer,

Thank you for taking the time to review our submission. In response to your comment please note the following:

  • We appreciate the comment to accept in the present condition. We did add the Mascharak et. al. 2021 paper to enhance the review.

We welcome any additional critiques to improve the review article.

Reviewer 3 Report

In the manuscript entitled “The Role of the Extracellular Matrix (ECM) in Wound Healing: A Review” the authors described the importance and the role of the extracellular matrix in the wound healing process. However, the contribute of this review to the current state of the art is residual, and the references used are outdated.

  1. The Introduction section of the manuscript is very confusing. Authors must rewrite it and make it more focused as well as update the content of it by using more recent references.
  2. Information concerning the wound dressings produced so far that are aimed to reproduce the ECM must be added to the manuscript.
  3. The authors must add a table to the manuscript, where the main roles of ECM must be summarized.

Author Response

Reviewer Three:

Dear Reviewer,

Thank you for taking the time to review our submission. In response to your comments please note the following:

  1. We have rewritten sections of the introduction and included an additional table that discusses commercially available products aimed at reproducing the ECM. Additional more recent references have been used.
  2. An ECM composition table has been added as a synopsis for the varying proteins/glycoproteins roles.

We welcome any additional critiques to improve the review article.

Round 2

Reviewer 1 Report

The manuscript has been improved after modification. 

I have still three comments:

1. You have to clarify the novelty of this review article. If there are other review articles with the same focus, then what would be the novelty of this work? Please clarify.

2. In a review article, you need to express your expert opinion before conclusion in a future perspective or outlook section, which is still missing.

3. References are improved after modification but still there are many too old and outdated references which have to be replaced by recent publications!

Author Response

Dear Reviewer,

Thank you for taking the time to review our submission. In response to your comments please note the following:

"1. You have to clarify the novelty of this review article. If there are other review articles with the same focus, then what would be the novelty of this work? Please clarify."

I have rewritten the final sentence in the abstract to help with this: This article synthesizes current and historical literature to demonstrate the involvement of the ECM in the varying phases of the wound healing cascade. 

"2. In a review article, you need to express your expert opinion before conclusion in a future perspective or outlook section, which is still missing."

I have added the following to the final paragraph of the manuscript:  This review addresses the role of the ECM throughout the wound healing cascade of events. Many wound healing products currently on the market in the US, have realized that using these structural proteins can increase the rate of wound heling in second intention wounds. Although this may not be enough. It is the opinion of the authors that more research needs to be devoted to not only utilizing the structural components of the ECM but to also mimic the physiological structure. 

"3. References are improved after modification but still there are many too old and outdated references which have to be replaced by recent publications!"

There are only a few dated references in our reference list, we are interested in referencing the source material that has led to understanding of the ECM, the dated references are not stand-alone references in most cases and are backed up with more recent publications. This also leads to a need for more research to conducted in the area of these dated references.

We welcome any additional critiques to improve the article.

Reviewer 3 Report

The input of this article to the current state of the art is very limited.

Author Response

Dear Reviewer,

Thank you for taking the time to review our submission. In response to your comment please note the following:

"1. The input of this article to the current state of the art is very limited."

I have rewritten the final sentence in the abstract to help with this: This article synthesizes current and historical literature to demonstrate the involvement of the ECM in the varying phases of the wound healing cascade. There are only a few dated references in our reference list, we are interested in referencing the source material that has led to understanding of the ECM, the dated references are not stand-alone references in most cases and are backed up with more recent publications. This also leads to a need for more research to conducted in the area of these dated references.

We welcome any additional critiques to improve the article.